Immersive virtual environments and embodied agents for e-learning applications

Fitton Isabel S. 1 isf21@bath.ac.uk
http://orcid.org/0000-0003-1169-2842 Finnegan Daniel J. 2
http://orcid.org/0000-0003-4066-3645 Proulx Michael J. 3
1 Department of Computer Science, University of Bath , Bath , UK
2 School of Computer Science & Informatics, Cardiff University , Cardiff , UK
3 Department of Psychology, University of Bath , Bath , UK
Ventura Sebastian
Electronic publication date: 2020 Nov 16
Publication date: 2020
Volume: 6
Electronic Location ID: e315
Received 2020 Apr 16; Accepted 2020 Oct 19
Copyright: © 2020 Fitton et al.
Copyright year: 2020
Copyright holder: Fitton et al.
License: This is an open access article distributed under the terms of the Creative Commons Attribution License, which permits unrestricted use, distribution, reproduction and adaptation in any medium and for any purpose provided that it is properly attributed. For attribution, the original author(s), title, publication source (PeerJ Computer Science) and either DOI or URL of the article must be cited.
License URL: https://creativecommons.org/licenses/by/4.0/

Keywords: Virtual reality, Distance learning, MOOC, Classroom, Immersive virtual environments, E-learning, Education

Funding: UKRI EPSRC Centres EP/L016540/1 and EP/T014865/1 School of Computer Science & Informatics at Cardiff University REal and Virtual Environments Augmentation Labs (ReVEaL) Research Centre Isabel S. Fitton’s research is supported in part by the UKRI EPSRC Centre for Doctoral Training in Digital Entertainment (CDE), EP/L016540/1. Daniel J. Finnegan thanks the School of Computer Science & Informatics at Cardiff University for their continued support. Michael J. Proulx is a member of the REal and Virtual Environments Augmentation Labs (ReVEaL) Research Centre, the UKRI Centre for the Analysis of Motion, Entertainment Research and Applications 2.0 (EP/T014865/1), and the UKRI Centre for Doctoral Training in ART-AI in Computer Science at the University of Bath. The funders had no role in study design, data collection and analysis, decision to publish, or preparation of the manuscript.

==============================
Massive Open Online Courses are a dominant force in remote-learning yet suffer from persisting problems stemming from lack of commitment and low completion rates. In this initial study we investigate how the use of immersive virtual environments for Power-Point based informational learning may benefit learners and mimic traditional lectures successfully. We examine the role of embodied agent tutors which are frequently implemented within virtual learning environments. We find similar performance on a bespoke knowledge test and metrics for motivation, satisfaction, and engagement by learners in both real and virtual environments, regardless of embodied agent tutor presence. Our results raise questions regarding the viability of using virtual environments for remote-learning paradigms, and we emphasise the need for further investigation to inform the design of effective remote-learning applications.

Introduction

Technological advancements have played a vital role in accommodating vast numbers of students through the growth of distance learning applications and e-learning platforms (Kaplan & Haenlein, 2016; Kauffman, 2015). The predominant form of distance learning applications are Massive Open Online Courses (MOOCs). MOOCs offer access to teaching and material on a large scale via internet-based virtual learning environments for a limitless number of participants, making education more accessible (Freitas & Paredes, 2018). Modern MOOCs involve a video captured recording of a human lecturer who delivers the learning content, facilitating the completion of homework or exams, and discussion via forums (Feng et al., 2015). However, despite the potential of MOOCs to deliver teaching materials and content at a global scale, existing platforms suffer from issues with drop-out and learner motivation (Yang et al., 2013). In parallel to e-learning platforms gaining popularity (Sneddon et al., 2018), VR technology has increasingly been adopted in the classroom as a teaching aid for ‘hands-on’ skills-based teaching partly due to reductions in cost. For example in medicine, digital models are much cheaper compared to physical anatomical models for training students (Rajeswaran et al., 2018). Using digital models in a virtual reality scenario is a cost-effective way to educate students on a large scale and as a result there is growing excitement regarding the potential of VR to revolutionise education and e-learning (Greenwald et al., 2017).

While VR is regarded as beneficial to students as a practical teaching aid, its application to formal, lecture style teaching—which e-learning platforms tend to deliver—is less common (Korallo, 2010). The use of Immersive Virtual Environments (IVEs) for corporate and higher-education purposes have only recently begun to emerge. Due to such applications being in their infancy there is very little empirical evaluation of their efficacy or research available to inform their design. A key component of many IVEs is the presence of an Embodied Agent (EA) which, in the context of learning, may serve as a virtual guide or tutor. The use of EAs as virtual tutors within educational IVEs is critical for effective pedagogy (Soliman & Guetl, 2010). Previous research suggests that the representation of artificial agents affects learners’ motivation (Maldonado & Nass, 2007). For example an EA may be customised by the learner to suit their preference—such customisation has been shown to improve performance for some cognitive tasks (Lin et al., 2017). In another study that tested male vs. female pedagogical agents, the female seemed to be preferred overall (Novick et al., 2019). A recent systematic review of pedagogical agents noted that positive results have been found in numerous studies, yet different combinations of features and different outcome variables have not been systematically studied to clarify which features work best or when (Martha & Santoso, 2019). However, it is unclear how the presence of an EA and learner motivation interact. A clear and robust analysis of these factors and their impact on the learning experience is critical for future application of IVEs as engaging platforms for distance learning.

Furthermore, the recent novel SARS-COV-2 (COVID-19) pandemic has had huge repercussions for higher education across the world. As millions of people were restricted to not leaving their homes for extended periods of time, many institutions also shifted to remote delivery of learning material for the academic year 2020/21. Although this presents challenges around blended learning and flipped classroom design, our focus remains on technology and how it may act as a medium for learning material delivery as opposed to what content such material should contain.

Our main contribution in this work is an empirical investigation into factors which impact the overall student experience when learning in IVEs. Specifically we report how the presence of an embodied teacher and students’ sense of presence in the environment impact learning retention, satisfaction and engagement, and student motivation to engage with learning material presented in an IVE. Our results demonstrate how learning in IVEs is comparable to real classroom learning, yet can scale far beyond the limits of traditional classrooms with constraints such as staff-student ratio and classroom size. Finally, we emphasise implications for future work in designing and assessing IVEs for remote learning purposes.

Background

As distance learning continues to expand, catering for larger numbers of students across the globe, current solutions provide inefficient delivery systems which are not immersive, engaging, or motivating to the learner—often resulting in poor rates of completion (Wise et al., 2004; Yang et al., 2013; Chen, 2018). For example an investigation into the use of ‘Accountable Talk: Conversation that Works’, a MOOC provided by the University of Pittsburgh, revealed that despite in excess of 60,000 students registering for the programme less than half continued to access the course material through to completion (Rosé et al., 2014). Attrition rates of learners in MOOCs is much higher than that in formal education (Clow, 2013; Joo, So & Kim, 2018). Users also report that they do not perceive MOOCs as equivalent to traditional education, and their engagement with them is less serious (Nemer & O’Neill, 2019). As a result, MOOCs are unable to deliver educational experiences with the same rigour as formal educational institutions. While MOOCs have several short-comings, the ability to study without being physically located in a certain space has many advantages to both students unable to attend and universities who are coping with growing numbers of students. Therefore, finding ways to improve the experience of distance learning and encouraging greater levels of engagement with online courses is of great public interest and their efficacy in education is of equal pedagogic interest.

Bringing learners into IVEs may overcome engagement issues experienced in MOOCs. Students prefer to engage in traditional lectures over online courses because they lack self-discipline and they can become too easily distracted during online learning (Crook & Schofield, 2017). Applying immersive VR to education may engage students better than MOOCs, removing distractions outside of the learning environment, mimicking the experience of traditional learning experiences (Lessick & Kraft, 2017; Pirker et al., 2018). Existing examples of educational applications of VR have focused on non immersive desktop-VR and have shown that simulating learning environments is highly effective. For example desktop-VR has been successfully used for social cognition training in children with Autism Spectrum Disorders and for assessing procedural skills such as dissecting frogs in a laboratory study (Didehbani et al., 2016; Merchant et al., 2014). IVE based learning environments have shown that learning using VR results in better retention and improves learners’ performance by up to a grade compared to simply watching a lecture or reading (Sitzmann, 2011; Graesser et al., 2005).

While educational applications of desktop-VR have merit, research suggests IVEs lead to better results as interaction with the environment is more intuitive, therefore users spend less time learning how to use the computer interface and can focus their full attention on the task (Psotka, 1995). To date, IVEs have been predominantly applied to procedural and skills-based education, successfully enhancing learning outcomes. For example the performance of a group of material science students on a series of questions about crystal structures improved when they were presented with virtual 3D diagrams of crystal structures via a head-mounted display (HMD), compared to using 2D diagrams (Caro et al., 2018). The ability to manipulate and rotate the crystals in the IVE helped students understand the relationships between atoms and perform better in assessment tasks than those who studied the textbook diagrams. Additionally, students reported that the IVE was easy to use and preferable over the 2D format. IVEs are also commonly used successfully to train complex psychomotor skills required by medical students. For example AirwayVR provides a safe, immersive environment to practice endotracheal intubation procedures, leading to clear improvements in students’ self-reported understanding of the procedure compared to their knowledge prior to using the application (Rajeswaran et al., 2018).

Recent research suggests applying IVEs to lecture-styled learning may provide distance learners with enriched learning experiences that are more immersive, enjoyable, and realistic (Chen, 2018). Preliminary research has shown the value in using IVEs to replicate classroom learning, finding that students perform better on a quiz about the topic after watching a virtual lecture compared to watching a video recording of a lecture, as is typically done in MOOCs (Tsaramirsis et al., 2016). Additionally, all learners reported that they preferred the IVE as it was more enjoyable, reinforcing the idea that IVEs are likely to successfully engage a larger number of distance learners than MOOC platforms. However, while educational applications of IVEs for distance learning in both higher education and corporate level training have begun to emerge, these applications are in their infancy. Recent work has explored the use of modern game development engines and HMD based environments for creating virtual lecture theatres and classrooms (Misbhauddin, 2018), but has not explored how effective these environments are at improving learner performance, motivation, and satisfaction & engagement. As such, to the best of our knowledge there are currently no published findings regarding their effectiveness, resulting in very little robust evidence to inform how the design of an IVE impacts learning outcomes (Moro, Stromberga & Stirling, 2017).

When designing IVEs, one does not consider just the aesthetic, but also the presence of other agents inside the environment. Within IVEs, embodied agents (EAs) are frequently used as pedagogical agents for virtual tutoring. For example STEVE is a human-like EA used to teach engineers how to use complex machinery onboard ships (Johnson & Rickel, 1997). In addition to humanoid EAs there are non-human examples such as Herman the bug—a non-humanoid EA implemented in Design-A-Plant, a virtual environment used to teach children about plant biology and the environment (Lester, Stone & Stelling, 1999). The appearance and behaviour of EA tutors influences learners’ feelings of co-presence (Baylor, 2011; Baylor & Kim, 2009)—the perception that one is not alone but in the presence of others (Heeter, 1992; Short, Williams & Christie, 1976). Co-presence increases when an EA tutor has appearance and behavioural realism—a key point being that there is no mismatch between appearance and behavioural realism, as this results in very low levels of perceived co-presence (Bailenson et al., 2005).

Increasing a learner’s perceived co-presence increases learner satisfaction and motivation to engage with material. For example it has been shown that learners spend approximately 25% more time learning and report that the learning experience is more enjoyable when an EA is present (Sträfling et al., 2010). A limitation of current distance learning platforms, such as MOOCs, is that learners must try to maintain enthusiasm and motivation to complete the course in the absence of an educator (Hasegawa, Uğurlu & Sakuta, 2014). Implementing an appropriate EA which represents a lecturer within an IVE may maintain learner interest and motivation, positively impacting learning outcomes. The appearance of the virtual tutor impacts a learner’s perception of the tutor’s abilities. For example human-like agents are perceived as more intelligent and helpful compared to non-human agents (King & Ohya, 1996; Lester & Stone, 1997), while familiar agents are rated more positively than unfamiliar agents (Bailenson et al., 2005). Previous research has demonstrated that virtual tutor realism influences learners’ reported likability and motivation (Maldonado & Nass, 2007), in turn influencing performance. Thus, we expect that a realistic EA tutor which is familiar to the learner will be more likeable, improving the learning experience and motivation to learn (Maldonado & Nass, 2007; Scaife & Rogers, 2001).

While some evidence suggests that EAs play a substantial role in the learning experience, increasing learning efficiency and retention (Roussou, Oliver & Slater, 2006), others have found minimal-to-no effect of EAs on learning outcomes. For example in one study EAs were found to have no influence and prior knowledge was identified as the greatest contributing factor to learner performance (Sträfling et al., 2010). Therefore, to clearly establish the utility of EAs within IVEs researchers should aim to control this potentially extraneous variable to prevent participants’ prior knowledge of the topic concealing any effects of the EA.

Overall, IVEs for educational purposes have the potential to mimic traditional learning experiences greater than MOOCs. By utilising IVEs to develop more engaging distance learning experiences, universities and corporate training bodies may cater for increasing student numbers. However, a major barrier to implementing IVEs compared to MOOCs is the higher relative cost of the equipment required. Therefore, it is essential that interdisciplinary research is conducted to establish whether IVEs, which can be run on low powered hardware such as smartphones, are able to provide a method of engaging more students, provide remote-learners with an experience which is more equivalent to formal education, and make a worthwhile contribution to higher education institutions looking to provide effective distance learning.

User study

Our study focuses on the educational applications of IVEs, specifically investigating the effectiveness of learning novel information in an IVE compared to a physical classroom, and the role of EAs as tutors within the IVEs. We devised the following hypotheses:

H1: Participants who learn inside an IVE will learn more effectively and outperform participants who learn in a physical classroom since prior research has shown that virtual learning environments result in better retention and improves performance (Sitzmann, 2011; Graesser et al., 2005).

H2: Participants who learn in the presence of an EA tutor will outperform participants who learn without one because the presence of a virtual tutor influences motivation which may in turn influence performance (Sträfling et al., 2010).

H3: The presence of a humanoid EA tutor will be more likable and increase motivation in learners compared to an abstract EA tutor because more familiar and realistic tutors are more likeable and motivating (Bailenson et al., 2005; Maldonado & Nass, 2007).

Materials and Methods

A between-participant design was used, whereby participants were randomly assigned to one of the four learning conditions. The independent variable was the learning environment (IVE with no tutor, IVE with a humanoid tutor, IVE with an abstract tutor, Non-virtual learning environment). The dependent variables were performance (test score), and reported motivation, satisfaction, and engagement (questionnaire). The experiment lasted approximately 45∼60 min.

Design and apparatus

Opportunity sampling was used to recruit participants from a local university. The target sample size for this initial study was 48 participants, split equally between the four learning conditions, based on the results of an a-priori power analysis, conducted using G*Power 3 (Faul et al., 2007), revealing a one-way ANOVA with 12 participants per group would provide 80% power to detect an effect size of 0.5 at a significance level of 0.05. Note that we instead used a more conservative Kruskal–Wallis test rather than the ANOVA to evaluate the results; prior work has shown that Kruskal–Wallis has greater statistical power than ANOVA under these conditions, so the analyses presented were indeed sufficiently powered (Hecke, 2012). In total, 48 participants were recruited (24 M, 24 F), aged 18 and over (M = 20.8 years, SD = 3.3 years). All participants were students from a variety of disciplines who reported normal or corrected to normal vision and hearing. Participants were incentivized through course credit (n = 8), £5 reward (n = 22), or simply volunteered to participate (n = 18). Statistical tests confirmed that there was no significant effect of the type of incentive received on participant performance (See Supplemental Material).

A machine running Windows 10 with a single Nvidia 970 GPU was adequate to drive the virtual environment since it is without cutting-edge graphics. To display the virtual environment, we used the HTC Vive HMD. This HMD covers a 110 degrees field of view, with two 1,080 × 1,200 pixel screens to render stereoscopic graphics to the viewer. Head position and orientation were tracked using the hardware base stations packaged with the HMD. However, the environment can also be demoed as a mobile application and we expect that in a larger cohort the set up could be easily scaled up using more consumer-friendly devices such as Smartphones and Google cardboards.

Learning environments & material

A seminar room on a local university campus was used as the non-virtual learning environment (See Fig. 1). A PowerPoint presentation was projected onto the screen to display the learning material and the female experimenter represented the tutor, reading a script alongside each slide (See Supplemental Materials).

Figure 1 The humanoid embodied agent tutor (A), the non-human tutor (B), a view of the entire virtual classroom environment (C), a view from the perspective of participants in our experiment showing the novel learning material (D), and a view of the real world classroom (E).

Participants sat in the same position in both real and virtual environments as shown by the red circles.

A virtual replica of the physical classroom, made to scale in order to minimise the number of extraneous variables (Fig. 1C) was created using Unity 2018.2.17. To replicate the appearance, colours and textures were applied and generic classroom furniture were used to decorate the virtual environment. To display the PowerPoint slides in the IVEs, custom software applied images of the PowerPoint slides as textures to the virtual projector screen. The lecture slide changed to the next one in sequence when spacebar was pressed. Audio recordings of the female experimenter reading the script were automatically played with each slide to keep delivery of the lecture material consistent for all participants.

For the IVE with a humanoid tutor, a female avatar was created using AdobeFuse CC Beta and imported into the environment (Fig. 1A). The female avatar has an animator controller to loop an ‘idle’ and an ‘eye blink’ motion to appear more realistic. For the IVE with an abstract tutor, a block-shape representation was created from geometrically primitive shapes (Fig. 1B). The abstract tutor was animated using key frame animation which moved the body side-to-side and rotated the eyes to replicate the humanoid ‘idle’ and ‘blink’ motions. Novel information was created for this study about the developmental stages of a made-up alien species. This was used as the learning material in order to eliminate the possibility of prior knowledge becoming a confounding variable (See Supplemental Material).

Questionnaire

All data were recorded using Qualtrics, a web browser interface that automatically recorded responses from participants. The first section of the questionnaire contained the knowledge test, composed of 29 questions designed to test participants’ knowledge of the alien species. The majority of the questions were multiple choice in order to test retention, with some short answer questions to test comprehension (Schrader & Bastiaens, 2012) (See Supplemental Material). However, multiple choice tests have been critiqued as they ‘feed’ students the answers, making it possible to gain artificially high scores (Bush, 2001). Therefore to accurately reflect retention, the test was negatively marked (meaning correct answers were given a score of 1, incorrect answers scored −1, and any unanswered questions scored 0) to discourage guessing (Davies, 2002). The test was marked to produce a score to indicate participant performance. Three blocks of questions followed: learner satisfaction and engagement (7 items); learner motivation (3 items); and virtual presence (5 items; see Supplemental Materials for the questionnaire).

Learner satisfaction and engagement

The questions designed to measure satisfaction and engagement with the learning experience were 5-point Likert scales which asked participants how strongly they agreed or disagreed with the statements. For example “The learning experience captured my interest”. Each item was scored out of five (5 = Strongly Agree) and added together to produce a learner satisfaction and engagement score out of 35. The questions were created for the purpose of this experiment as it aimed to measure specifically how engaged ‘students’ were in this one experience. We had considered using an existing student satisfaction questionnaire but opted to develop our own so that questions could be focused on the experience in our study.

Learner motivation

Another block of questions was specifically tailored to investigate the effects of the EA tutor manipulation on learner motivation, for example “The presence of the tutor increased my motivation to learn”. Each item included a 5-point Likert scale which asked participants to what extent they agreed with each statement, with ‘Strongly Agree’ being scored as five. The item scores were added together to produce a learner motivation score out of 15.

Virtual presence

The virtual presence questions were taken from an existing questionnaire (Witmer & Singer, 1998). The most applicable items were selected, for example “To what degree did your experiences in the virtual environment seem consistent with your real-world experiences?”, participants responded to each statement via a 5-point scale (1 = Not at all, 5 = Completely) to indicate how immersive the IVEs were, and this produced a virtual presence score out of 25.

Finally, to measure any potentially confounding effects, participants in the IVE with a humanoid tutor were asked to report anything they found ‘odd’ about the human avatar, as a perceived mismatch between appearance and expected behaviour, for example ‘speaking’ with no changing facial expression or lip movement, could lead to disliking of the tutor and affect performance (Mori, 1970; Bailenson et al., 2005).

Procedure

This study was approved by the University of Bath Psychology Research Ethics Committee (17-292). Participants were provided with further information about the study before giving written informed consent. Only one participant took part in the experiment at a time. Each participant was randomly allocated to one of the four learning conditions.

Participants in the IVEs were seated at a computer in the laboratory, the experimenter would assist with fitting the headset and headphones to ensure the participant was comfortable. The experimenter would then launch the appropriate classroom application (i.e. with a humanoid/abstract/no tutor). Participants allocated to the non-virtual condition were seated in a seminar room on the university campus. Participants in the non-virtual condition took part in the experiment individually: the only other person present in the room was the experimenter. In both virtual and non-virtual conditions participants were shown the same PowerPoint presentation, and heard the same experimenter deliver the scripted information. The only difference being that in the virtual condition the voice-clips were pre-recorded and incorporated into the environment, whereas in the non-virtual condition the experimenter delivered the information in person. In all conditions, participants observed the full presentation with corresponding audio once, and were then allowed the remaining time to read through the slides themselves with no audio input. After 30-minutes the experimenter halted the learning part of the experiment, and those in the IVEs would be asked to remove the headset. All participants were then required to complete the online test and questionnaire.

Throughout the experiment, all participants remained naïve to the manipulation of the tutor and the environment. Afterwards all participants were fully debriefed, and the full aims of the study were revealed, participants then provided final consent for the data to be used.

Analysis and Results

Frequentist null hypothesis significance testing and the associated p-value has many shortcomings, for example it relies on hypothetical data and can be easily manipulated—with larger sample sizes able to make small differences significant without any practical value (Jarosz & Wiley, 2014; Dienes, 2011). Bayes Factor (BF) is a ratio which indicates the likelihood of the observed data fitting under either of the two hypotheses. Therefore, Bayesian statistics were conducted using JASP version 0.9 to determine the relative strength of the support for the null vs. alternative hypotheses. BF represents the likelihood that the evidence is explained by one hypothesis over another, for example a BF of 20 would indicate that one hypothesis is 20 times more likely to explain the data. BF can be given as BF10 (evidence for the alternative hypothesis) or BF01 (evidence for the null hypothesis) (Schut et al., 2018). We used BF01 values as they are easier to interpret in relation to our findings. Based on this interpretation scheme, BF01 values of 3–10 indicate moderate support for the null hypothesis, while values <3 indicate weak support for the null hypothesis (Wagenmakers et al., 2018). Tables 1 and 2 presents a summary of these results.

Table 1 Kruskal–Wallis results summary covering the three core variables in our study: learner performance, satisfaction & engagement, and sense of presence.

	Non-virtual	Virtual, no tutor	Virtual, humanoid tutor	Virtual, abstract tutor	H(3)	p	η2	
Measure	M	SD	M	SD	M	SD	M	SD	
Performance	30.75	11.67	28.92	10.22	28.33	9.20	26.33	8.17	2.806	0.422	0.06	
	no tutor	humanoid tutor	abstract tutor			H(2)	p	η2	
	M	SD	M	SD	M	SD			
Satisfaction & Engagement	27.42	5.35	25.75	5.91	25.67	5.77			2.954	0.399	0.063	
Presence	14.42	2.27	14.67	2.46	14.42	2.39			0.066	0.968	0.001	

Table 2 t-test results for the impact of tutor on motivation to learn.

	Abstract tutor	Humanoid tutor				
Measure	M	SD	M	SD	t(22)	p	d	
Motivation	10	2.2	8.8	2.1	−1.316	0.202	−0.537	

Additional statistical analyses carried out using SPSS version 24.0 were evaluated against an alpha level of 0.05. An independent t-test was used to determine differences in learner motivation between the humanoid tutor IVE and the abstract tutor IVE. Assumption checking revealed that the data were normally distributed as assessed by the Shapiro–Wilk test (p > 0.05), and the variance in each group was approximately equal, as assessed by Levene’s test for homogeneity of variances (p > 0.05). However, due to the small sample size three Kruskal–Wallis tests were conducted to compare learner performance, satisfaction and engagement, and virtual presence ratings across multiple learning conditions. Preliminary analyses confirmed that the data met the test assumptions as there were no extreme outliers, and there was homogeneity of variances.

Learner performance

Mean scores for learner performance in each learning conditions are shown in Fig. 2. On average, learner performance was similar across all learning environments, with only slight differences among conditions. With respect to H1 and H2, we conducted a Kruskal– Wallis test and results report no statistically significant differences in performance scores between conditions, H(3) = 2.806, p = 0.422, η2 = 0.06, this result is moderately reinforced by the Bayes statistics which indicate that the data are six times more likely to be explained by the null hypothesis (BF01 = 6.2).

Figure 2 Mean test scores in the different learning environments. Error bars represent the standard error (SE).

Dots show distribution of participant scores, with larger dots indicating multiple participants with the same score.

Learner satisfaction and engagement

Mean scores for learner satisfaction and engagement in each virtual learning condition are shown in Fig. 3. Learner satisfaction and engagement levels, as measured by seven items in the questionnaire (Cronbach a = 0.84), were similar across all virtual learning conditions, with only slightly higher levels measured in the IVE with no tutor. A Kruskal–Wallis test supported that learner satisfaction and engagement levels were not significantly different between the virtual learning conditions, H(3) = 2.954, p = 0.399, η2 = 0.063 Furthermore, Bayes statistics indicate that the data are four times more likely to be explained by the null hypothesis (BF01 = 4.1).

Figure 3 Mean satisfaction and engagement scores with error bars representing SE, for the predictor of learning condition within immersive virtual environments (no tutor, humanoid tutor, abstract tutor).

The real environment was not modelled as a condition in this analysis and therefore means for three conditions are shown. Error bars and dots as in Fig. 2.

Learner motivation

Mean scores for learner motivation in the presence of humanoid and non-humanoid EAs are shown in Fig. 4. Learner motivation scores, measured using three items in the questionnaire (Cronbach a = 0.62), appeared higher in the IVE with the abstract tutor (M = 10.0, SD = 2.2) compared to learner motivation scores in the IVE with the humanoid tutor (M = 8.8, SD = 2.1). With respect to H3, an independent samples t-test revealed that these differences in learner motivation between conditions were not significantly different, t(22) = −1.316, p = 0.202, d = −0.537. However, Bayes statistics only indicate very weak support for the null hypothesis in this case as it suggests that the null hypothesis is only one times more likely to explain the data (BF01 = 1.4).

Figure 4 Results of a t-test conducted to analyse the impact of humanoid vs. abstract tutor representation on learner motivation scores, and therefore means for two conditions are shown.

Error bars and dots as in Fig. 2.

Virtual presence

Virtual presence was measured using only a subset of Witmer and Singer’s presence questionnaire so measures of internal reliability were not conducted. Learner ratings of virtual presence were consistent across the different IVEs (no tutor M = 14.4, SD = 2.3; human tutor M = 14.7, SD = 2.5; abstract tutor M = 14.3, SD = 2.3). A Kruskal–Wallis test supported that virtual presence scores were not significantly different between the virtual reality learning conditions, H(2) = 0.066, p = 0.968, η2 = 0.001, furthermore Bayes statistics provide moderate support for the null (BF01 = 5).

Learner perceptions of avatar

In response to the question “Did you notice anything odd about the human avatar?” 58% of the participants in the IVE with the humanoid tutor reported that they did find the humanoid EA tutor strange. The reasons for answering ‘yes’ to the question were that the avatar had strange or repetitive movement, no changing facial expressions, and did not speak.

Discussion

Previous research and educational applications of VR have focused on desktop-VR simulations for skills-based tasks (Freina & Ott, 2015), neglecting the use of more IVEs and their potential use for informational, lecture-styled learning experiences. Therefore, in this pilot study we investigated to what extent informational-learning within an IVE is effective compared to learning in a physical classroom. We created a virtual replica of a classroom and compared its use for informational learning to the traditional, real-world classroom.

Previous literature indicated that simulated learning environments are highly effective, enhance declarative knowledge, and lead to better retention compared to conventional learning methods (Graesser et al., 2005; Merchant et al., 2014; Sitzmann, 2011). While desktop-VR has dominated the literature, it is argued that more immersive experiences may lead to even greater results (Psotka, 1995). Therefore, we hypothesised that participants who learned virtually would outperform participants who learned non-virtually (H1). However, results demonstrated that participants who learned virtually did not outperform participants who learned non-virtually. A BF analysis provides moderate support for this finding as it suggests that the data are six times more likely to be explained by the null hypothesis. It is plausible that familiarity with learning material, which is known to impact learner engagement, performance, and motivation (Schönwetter, Clifton & Perry, 2002), would have impacted our results: to combat this effect we used fabricated information to eliminate prior knowledge as a confounding variable. Thus our findings are robust, indicating there is no detriment to learning in an IVE compared to a conventional classroom setting (Madden et al., 2020).

The lack of a statistically significant difference in learner performance between the IVE and the non-virtual classroom is of particular importance as educational institutions are under increasing pressure to cater for large numbers of students, and as such require effective distance learning applications (Kauffman, 2015). Current distance learning platforms suffer from poor student engagement and high levels of drop-out (Yang et al., 2013), however, IVEs have the potential to improve this. Previous research has demonstrated that IVEs provide a more engaging platform for distance learners than existing video-based applications (Tsaramirsis et al., 2016). In light of the COVID-19 pandemic and its impact on the higher education sector, namely creating situations where many institutions have closed their campus until further notice, IVEs may yield a better experience for distance learners. Furthermore, our research supports that distance learning applications would benefit from incorporating the use of IVEs, as distance-learner performance would be consistent with learners in traditional settings, yet IVEs can be used on a much larger scale, making them highly cost-efficient.

The role of embodied agents

Few studies have been conducted which inform how the design and use of EA tutors impacts the learner experience (Moro, Stromberga & Stirling, 2017), so we investigated the role of EA tutors within IVEs. We manipulated the tutor in the IVE on two dimensions; its presence or absence, and its human-like representation. Previous research has suggested that co-presence influences the learning experience, with higher feelings of co-presence resulting in greater learning performance (Roussou, Oliver & Slater, 2006; Wise et al., 2004). Therefore, we hypothesised that participants who learn in the presence of an EA tutor will outperform participants who learn in its absence.

We found no statistically significant difference in performance of learners who learned without a virtual tutor, with a humanoid tutor, or with an abstract tutor. Participants who learned in the IVE without an EA tutor were expected to perform worse on the post-learning test. Our findings fit into current discourse and debate on the utility of EAs and their impact on learning: while some research has concluded no impact on performance, consistent with ours (Sträfling et al., 2010), other research has found that EAs do impact learning performance (Baylor Amy, 2009; Maldonado & Nass, 2007; Rosenberg-Kima et al., 2007). One explanation for our results is the age of participants in our study. Previous research in agreement with our results used young adults (Sträfling et al., 2010), while others have recruited young school children (Roussou, Oliver & Slater, 2006). The presence of virtual avatars is known to positively impact learning in young children (Darves, Oviatt & Coulston, 2002) and it is possible that the positive impact of EA tutors on learner performance may be confined to when IVEs are used by younger students (Baylor & Kim, 2004; Ashby Plant et al., 2009).

Embodied Agent tutors impact learner satisfaction and engagement within IVEs by simulating the relationship between student and tutor (Alseid & Rigas, 2010). A learner’s social judgement of interactions with an EA impacts perceived co-presence and satisfaction, with human-like representations regarded as more social than non-human avatars (Nowak, 2004). Therefore, we hypothesised that a humanoid EA tutor would be preferred over an abstract EA. However, our results show no statistically significant differences in learner satisfaction and engagement when comparing the IVE with no tutor, the humanoid tutor, or the abstract tutor. BF analysis indicates that the data were four times more likely to be explained by the null hypothesis in this instance and therefore can be accepted with moderate confidence (Wagenmakers et al., 2018).

Although measures were taken to provide the humanoid tutor with a realistic appearance and behaviour, such as using deictic gestures and a natural human voice (Atkinson, Mayer & Merrill, 2005; Baylor, 2011; Baylor & Ryu, 2003; Janse, 2002), the avatar was not equipped with any animations which replicated changing facial expressions. This may have hindered the level of satisfaction and engagement the humanoid tutor was able to evoke in the learners, which is known to influence perceived realism (Atkinson, 2002). In our study, many participants exposed to the humanoid tutor commented on the absence of facial expression, with the majority of participants feeling as though it had ‘strange’ and ‘repetitive movement’. In contrast, participants did not have pre-defined expectations of how the abstract tutor should behave and as such it was not susceptible to the uncanny valley effect, unlike the humanoid avatar (Bailenson et al., 2005; Mori, 1970). Therefore, rather than the humanoid tutor increasing motivation, participants may have found the abstract tutor more appealing. This mismatch between learner expectations of the tutor and reality may have been detrimental to the perceived co-presence (Bailenson et al., 2005). Therefore, it is possible that a lack of emotional expression in the humanoid avatar contributed to the absence of significantly greater learner satisfaction and engagement.

Previous research has indicated that EA tutors affect learning outcomes indirectly by influencing learner motivation (Baylor, 2011). Research suggests greater likeability, and ascribed intelligence when using a humanoid EA tutor (King & Ohya, 1996; Lester & Stone, 1997); therefore, we expected the humanoid tutor would increase levels of learner motivation. However, there was no statistically significant difference in learner motivation between the humanoid and abstract tutor groups immersed in the VLE. BF analysis suggests that the data are almost equally likely to be explained by either the null or the alternative hypothesis. Thus, we do not rule out the possibility that tutor appearance can affect motivation.

A distinct strength of our study is the control for immersion as a factor which could influence the efficacy of IVEs, as the more immersive the environment the more comparable it is thought to be to real-world environments. To determine if learning outcomes are affected by immersion a virtual-presence questionnaire was used. The results indicated similarly high levels of immersion in all three IVEs, meaning that the IVEs are comparable to non-virtual learning (Peperkorn, Diemer & Mühlberger, 2015; Shin, 2018) and ensuring that environment quality was unlikely to produce any differences in learning outcomes.

Future work

While this pilot study provides preliminary support for the use of IVEs by demonstrating that learning within an IVE is not significantly different to non-virtual learning, this is only demonstrated in the immediate short-term as the test and outcome measures were administered immediately after the learning experience. For effective distance learning applications, long-term outcomes need to be assessed, perhaps in the realm of a longitudinal study. Future work should consider incorporating additional follow-up assessment periods, in order to provide evidence for whether the performance outcomes observed in the IVE and the non-virtual classroom are maintained over a longer period of time. In this preliminary study recruitment was restricted to the student population at the university, however it is likely that large differences in learning styles and ability will vary within this population resulting in a large range of scores in all conditions. Future studies using a larger sample-size should consider the prior grades of all participants, and use random allocation to minimise the effects of individual differences.

Additionally, the present study highlights the need for further investigation into the impact of EAs in IVEs to understand the varying results surrounding their impact on learner performance, motivation, and satisfaction. Previous work has highlighted the impact of graphical realism on peoples’ perceptions of avatars in virtual environments while engaging in various tasks in various scenarios (Tessier et al., 2019; Kang, Watt & Ala, 2008; Lugrin et al., 2015). Our goal was to assess the importance of a humanoid avatar, not necessarily the physical realisation of said avatar. Future work may consider graphical fidelity and realism as a factor in learner motivation, presence, and satisfaction and engagement with the learning experience. A possible trend in the literature is based around the age of participants, with younger participants seemingly more likely to be influenced by the presence of an EA. Future research should seek to investigate this theory.

In the present study there was no verbal interaction allowed between participants and the tutor in all learning conditions in order to remove the likelihood of differing levels of social interaction between participants and the tutor becoming a confounding variable. Furthermore, the EA was not equipped with any facial animation to replicate changing expression, both of which likely had a negative impact on the perceived realism. Future work investigating learner satisfaction, engagement, and motivation should consider introducing EAs with changing expressions and allow verbal interaction, such as the ability to ask the tutor questions, as this may better simulate student-tutor relationships and have a greater impact upon perceived co-presence, producing more insightful results regarding the role of EA tutors within IVEs. It may also reduce the strangeness reported in “Discussion” as the avatar’s behaviour is improved.

Finally, we highlight a novel avenue for future research: whether the influence of an EA on learning outcomes are mediated by their relevance to the learning material itself. In our study, participants studied fabricated information about an alien species, hence the abstract tutor may be more salient in this context, promoting interest in the learning material to a greater extent than the humanoid tutor (Maldonado & Nass, 2007). Future work will assess the link between learning material and the EA tutor’s appearance, as well as its contextual relevance and form within the IVE.

Conclusions

To the best of our knowledge, this pilot study is the first to directly compare informational-learning in a traditional classroom to a virtual replica using immersive VR for groups of participants in a controlled, laboratory setting. Our findings suggest that learner performance is equivalent in both learning situations. It remains unclear how the design of the IVE might impact learning outcomes, in particular whether the presence and appearance of the virtual tutor plays a role in learning outcomes. We have discussed avenues for future work, building on our preliminary study and exploring other factors which may impact learning performance in IVEs as well as guidelines and recommendations for how to design future experiments which control for extraneous variables. There are important implications for developers of distance learning applications: by providing IVEs opposed to video-based applications, it is possible to reduce the issues with current distance learning platforms and achieve comparable performance levels with those who learn in a traditional classroom, making it possible to cater for increasing numbers of students. As academic and corporate education moves towards IVEs under increasing pressure to meet the demands of growing numbers of students (Kauffman, 2015), the scalability and significant financial incentives they provide while maintaining satisfactory learning outcomes make them an attractive alternative to current video-based distance learning platforms. In addition, the COVID-19 pandemic has also forced institutions to rethink their ability to provide effective blended learning and virtual learning environments for students. IVE technology can help to create effective learning environments that are safe for staff and students, and continue to provide high quality learning and teaching.

Supplemental Information

Supplemental Information 1 Raw participant data sheet.

Click here for additional data file.

Supplemental Information 2 Lecture slides containing the learning material presented to participants during the experiment.

Contains information about a made-up alien species which participants had to learn about during the study.

Click here for additional data file.

Supplemental Information 3 Study Questionnaire that was presented to participants online using Qualtrics.

The first section of the questionnaire contains the knowledge test, composed of 29 questions designed to test participants’ knowledge of the alien species. Three blocks of questions followed: learner satisfaction and engagement (7 items); learner motivation (3216items); and virtual presence (5 items).

Click here for additional data file.

Supplemental Information 4 Codebook: contains the values needed to convert numbers in the dataset to their respective factors.

Click here for additional data file.

Additional Information and Declarations

Competing Interests

Author Contributions

Ethics

Data Availability

The authors declare that they have no competing interests.

Isabel S. Fitton conceived and designed the experiments, performed the experiments, analysed the data, performed the computation work, prepared figures and/or tables, authored or reviewed drafts of the paper, and approved the final draft.

Daniel J. Finnegan conceived and designed the experiments, performed the computation work, prepared figures and/or tables, authored or reviewed drafts of the paper, and approved the final draft.

Michael J. Proulx conceived and designed the experiments, authored or reviewed drafts of the paper, and approved the final draft.

The following information was supplied relating to ethical approvals (i.e., approving body and any reference numbers):

The University of Bath Psychology Research Ethics Committee granted approval to carry out this study (Approval number 17-292).

The following information was supplied regarding data availability:

The raw data is available in the Supplemental Files.

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
