# Peer review of "Immersive virtual environments and embodied agents for e-learning applications"

_PeerJ Computer Science, doi:10.7717/peerj-cs.315_

## Round 0.1 · original submission · Major Revisions

Please attend all the reviewers' suggestions in the new version of the paper.

Reviewer 1 ·

Basic reporting

The use of English is correct. Literature references are enough and the context of the research is well described.

I recommend authors including one or more tables with some results of the study to improve manuscript legibility.

Each of the individual images of Figure 1 should be referenced throughout the text, maybe with a letter (a), (b), ... and properly referenced from the text.

Please, revise the capitalization of reference Sitzmann (2011).

There is a missing white space in Line 190

Should Figures 2, 3, and 4 show bars for the four learning environments?

Experimental design

The research method is described with detail and the manuscript is accompanied by the raw data for replication. The authors have the approval of an Ethics Committee.

The gender was one of the data collected from students, but there is no comment about that in the manuscript.

Please, elaborate on this sentence "existing student satisfaction questionnaires were too broad and inappropriate for this application." (line 224)

Validity of the findings

Conclusions are well stated and speculation is controlled in the text. However, the authors are comparing different VR environments in which the developed humanoid avatar is far from providing a good realistic experience. From my point of view, that is an important threat to the validity of the results.

The virtual experiences are individual whereas the non-virtual ones are in a group with other students. Could this fact affect the results?

The authors used fabricated information to eliminate the possibility of prior knowledge becoming a confounding variable. Could results change whether serious content was used instead? From my point of view, that is a factor authors should be studied for this present research.

Additional comments

I am not very convinced about the utility of the study which analyzes the use of immersive virtual environments for Power-Point based learning. From my point of view, this kind of passive learning doesn't require that virtuality. So, I don't appreciate any potential impact in the paper, but according to the journal guidelines, that is not a concern.

·

Basic reporting

The paper presents a study that now more than ever is very relevant. The new COVID situation is forcing universities to quickly shift to virtual learning and thus, it is very interesting to see the study results. (I would also add a paragraph to stress the importance of such practices now in the light of the pandemic).

Experimental design

I am pleased with the experimental design. I only have one question:
Line 177: some participants were paid and some did not. Didn’t this create uneven conditions?

Validity of the findings

With a small size like the one used in the study, normality assumptions are usually not met. In this case, you performed a normality test but I believe you should also use a Kruskal Wallis test. This non-parametric test will provide stricter results and the significance of your findings will be stronger. I would also suggest to run a Cronbach analysis to test the internal validity of your questionnaire.

Additional comments

This is a very well written work that is also vey relevant due to the current pandemic situation. I have suggested a couple of additions and I believe that the paper will be ready for publication.

Reviewer 3 ·

Basic reporting

The paper is clear and self-contained with relevant results to hypothesis set.
The paper is unambiguous, it is well written and it has a good structure.

The authors need to better survey the literature for recent and relevant works on the effectiveness of embodied pedagogical agents and their impact on students learning.

The work also needs a better connection with existing works and authors need to explain how it builds on the findings of related works

Experimental design

The experimental designed is good. I suggest authors to better explain the exact phases of the experimental study. Also the background of the participants need to be discussed in better detail.

The same stands for the results too. Authors need to elaborate and further discuss them.

Validity of the findings

The novelty of the study should be pointed out by the authors in the introduction. Also, the data that was collected via the participants in the experiment and the exact analyses that were performed need to be explained in better detail too.

Additional comments

Authors in this paper address a very challenging topic that of virtual pedagogical agents and examine their impact on students learning experience and motivation.
The paper is interesting and it has novelty. In general, I can say that it is well written, it has a good structure.
The introduction could be restructured in order to explain the exact aims of the paper, its highlight out more explicitly the contributions of the work. Also please explain how the present work in could be assistive to the related research community.
The paper is clear and has specific hypotheses that it set. The results are interest. Authors need to address the aforementioned comments in order to improve their work.

---

## Round 0.2 · accepted · Accept

Both reviewers consider the paper is already acceptable for publication, and so do I. Congratulations

Reviewer 1 ·

Basic reporting

The authors have properly addressed the posed questions and suggestions.

Experimental design

The authors have properly addressed the posed questions and suggestions.

Validity of the findings

The authors have properly addressed the posed questions and suggestions.

Additional comments

The authors have properly addressed the posed questions and suggestions.

·

Basic reporting

The article is well written and easy to follow. The reference list has ben enhanced.

Experimental design

The methodology is adequate and the authors have performed extra statistical analyses, as instructed by the reviewers

Validity of the findings

The statistical methods used provide reliable and valid results, especially in the second version that the authors also used non-parametric approaches.

Additional comments

The paper in now significantly improved and the authors have addressed all reviewer comments. I will suggest publication.